# 90K/Mac-2 BP Is a New Predictive Biomarker of Response to Infliximab Therapy in IBD Patients

**DOI:** 10.3390/ijms24043955

**Published:** 2023-02-16

**Authors:** Pasqua Letizia Pesole, Marina Liso, Rossella Donghia, Vito Guerra, Antonio Lippolis, Mauro Mastronardi, Palma Aurelia Iacovazzi

**Affiliations:** National Institute of Gastroenterology—IRCCS, “Saverio de Bellis”, 70013 Castellana Grotte, Italy

**Keywords:** inflammatory bowel disease, 90K/Mac-2 BP, galectin-3 binding protein, infliximab, biological drug, biomarkers, Crohn’s disease, ulcerative colitis

## Abstract

Inflammatory bowel diseases (IBD), comprising Crohn’s disease (CD) and Ulcerative Colitis (UC), are multifactorial disorders characterized by a chronic inflammatory status with the secretion of cytokines and immune mediators. Biologic drugs targeting pro-inflammatory cytokines, such as infliximab, are broadly used in the treatment of IBD patients, but some patients lose responsiveness after an initial success. The research into new biomarkers is crucial for advancing personalized therapies and monitoring the response to biologics. The aim of this single center, observational study is to analyze the relationship between serum levels of 90K/Mac-2 BP and the response to infliximab, in a cohort of 48 IBD patients (30 CD and 18 UC), enrolled from February 2017 to December 2018. In our IBD cohort, high 90K serum levels were found at baseline in patients who then developed anti-infliximab antibodies at the fifth infusion (22 weeks after the first), becoming non-responders (9.76 ± 4.65 µg/mL compared to 6.53 ± 3.29 µg/mL in responder patients, *p* = 0.005). This difference was significant in the total cohort and in CD, but not significant in UC. We then analyzed the relationship between serum levels of 90K, C-reactive protein (CRP), and Fecal calprotectin. A significant positive correlation was found at baseline between 90K and CRP, the most common serum inflammation marker (R = 0.42, *p* = 0.0032). We concluded that circulating 90K could be considered a new non-invasive biomarker for monitoring the response to infliximab. Furthermore, 90K serum level determination, before the first infliximab infusion, in association with other inflammatory markers such as CRP, could assist in the choice of biologics for the treatment of IBD patients, thereby obviating the need for a drug switch due to loss of response, and so improving clinical practice and patient care.

## 1. Introduction

Inflammatory Bowel Diseases (IBD) include Ulcerative Colitis (UC) [1] and Crohn’s Disease (CD) [2], both characterized by alternating phases of exacerbation of symptoms and clinical remission [3]. The pathogenesis of IBD is not entirely clear, but we know that, in genetically predisposed individuals, it can be triggered by one or more factors acting in a vicious circle, self-fueling the chronic inflammation. Among these: various inappropriate and deregulated activations of the immune system [4,5,6,7,8]; several environmental factors; diet and intestinal dysbiosis, a condition characterized by an imbalance in the gut microbial community, thus favoring the prevalence of pathogens versus healthy microbial species [9,10,11]. Currently, the main target of IBD therapies is tumor necrosis factor (TNF), a pro-inflammatory cytokine. Anti-TNF drugs, which include infliximab (IFX), Adalimumab, Golimumab, and Certolizumab pegol, have been introduced into clinical practice for treating IBD [12,13,14,15]. Among these, infliximab, a chimeric anti-TNF monoclonal antibody, is crucial to reducing the need for surgery, improving the quality of life, and cutting down hospitalization in patients [12,13,14,15]. This biological drug is an anti-TNF agent that has completely revolutionized the pharmacological therapy of IBD and rheumatoid arthritis patients. Nevertheless, patients are often unresponsive to the IFX infusion, or become secondary non-responders at subsequent infusions [13,16,17,18]. Lowering circulating levels of the drug, increasing drug clearance, and the development of antibodies to infliximab (ATIs), are determinants of the loss of response to IFX therapy [19,20,21,22,23]. Therefore, monitoring the levels of the drug or ATIs may guide therapeutic decisions in patients with either clinically suspected or manifest loss of response to the therapy [19].

In particular, therapeutic drug monitoring (TDM) has been proven to maintain the efficacy of biologic agents, prevent immunogenicity, and hence improve clinical results and reduce costs in IBD patients management [24,25,26,27,28].

Recent data have demonstrated that proactive TDM was associated with an improvement in clinical remission and a reduction in treatment failure and hospitalization, while reactive TDM was considered to be more effective as compared with empiric dose escalation, and for this reason is recommended by the majority of gastroenterology associations [29,30,31,32]. Moreover, research into non-invasive biomarkers, in association with TDM, is important for clinicians in order to advance personalized therapies with the aim of controlling inflammation or, even better, inducing disease remission.

C-reactive protein (CRP) is the most common serum biomarker of inflammation in IBD, and its characteristics are useful for grading inflammation, monitoring the response to therapy, and to identify recurrent inflammation after medically or surgically induced remission [20,32]. Recent studies associated CRP levels with biomarkers and clinical disease activity indices in IBD patients [33,34,35,36,37]. In two distinct cohorts of pediatric IBD patients, an elevated CRP level was considered a risk factor for biologics treatment [34,35]. In a multicenter study, the CRP levels were used to monitor the clinical response between the anti-TNF drug Adalimumab and biosimilars in IBD patients, both CD and UC; no significant difference was found in CRP levels between patients switching from the originator to biosimilar or between biosimilars [36].

Recently, we demonstrated that a high level of circulating Interleukin-1β (IL-1β) is predictive of non-response to infliximab, in a subpopulation of UC patients [38].

90K/Mac-2 BP (Mac-2-binding protein), also known as 90K, lectin galactoside-binding soluble 3-binding protein (LGALS3BP) or Galectin-3 binding protein (Gal-3BP), is a N-glycoprotein (PM 70–95 kDa), originally described as a circulating antigen in the supernatant of human breast cancer cells [39]. It belongs to the Scavenger Receptor Cysteine Rich (SRCR) superfamily, which includes proteins most implicated in the defense functions of the host [40]. In our previous studies we found a progressive increase of 90K serum levels during the transformation from chronic hepatitis, through cirrhosis, to HCC [41,42]. The correlation between 90K and other cytokines’ concentrations in asymptomatic patients and healthy individuals also showed that 90K can be part of an immune and inflammatory response network, driving the activation of the host’s immune defense systems [43]. The physiological functions of extracellular protein include immunomodulation and the modulation of proliferation, motility, and migration and cell adhesion, through the interaction with cell-surface receptors. Specifically, the upregulation of cytokine production, such as IL-2, IL-6, granulocyte-macrophage colony-stimulating factor (GM-CSF), and TNF, is associated with the induction of 90K in several viral infections including SARS-CoV-2 [44,45,46,47]. Treatment with 90K of PBMCs induces the secretion of various cytokines such as IL 6, GM-CSF, and TNF-alpha [48].

In 2019, Cibor et al. found higher 90K serum levels in a group of patients with active CD compared to those in healthy controls, indicating a role for serum 90K as an adjuvant biomarker for disease activity, even if not a specific marker for CD [45]. However, they did not correlate circulating 90K with the response to therapy, as their CD patients were undergoing different therapies (corticosteroids, biologics, and immunomodulators), with no correlations among them.

Recent studies demonstrate that measuring infliximab trough levels (ITLs) in sera is useful for controlling the inflammation rate in IBD patients, without increasing the risk of infection [49,50,51,52].

The aim of this study is to analyze the relationship between 90K serum levels measured at baseline, together with circulating levels of IFX (ITLs) and ATIs, in IBD patients undergoing IFX administration, in order to clarify the role of serum 90K as an adjuvant biomarker for IBD patients in the active phase. Furthermore, we aim to evaluate the clinical usefulness of this new biomarker in the management of IBD patients undergoing IFX therapy.

## 2. Results

### 2.1. Baseline Characteristics

The clinical features, serological, and fecal data of the enrolled patients are shown in Table 1. In total, 48 IBD patients (30 CD and 18 UC) were enrolled on the day of the first IFX infusion (Appendix A); 47 patients completed the study and one dropout in the CD cohort was registered before the fifth infusion. Among the patients, 16.13% were smokers; none of the patients consumed excessive amounts of alcohol (>20 g/day).

All the UC patients had a moderate or severe disease, determined by the Mayo full score (Table 1); 26.32% of them had distal UC, and the remaining patients had pancolitis. In the CD patients, the disease behavior was inflammatory in 81.48% (B1), stricturing in 14.81% (B2), and penetrating in only 3.70% (B3); none of the CD patients had perianal disease. According to the Montreal Classification, 33.33% of the CD patients had ileal disease (L1), 11.11% a colic phenotype (L2), and 55.56% an ileocolic disease (L3); none of the patients had an upper-gastrointestinal location (L4). The mean HBI (Harvey–Bradshaw Index) measured for the CD patients was 15.52 ± 4.40, indicating a moderate grade of disease (Table 1).

Table 1 shows the baseline characteristics of the enrolled patients, stratified by disease. In UC patients we detected higher CRP levels and lower circulating iron, ferritin, and FCP (fecal calprotectin), compared to CD patients. These differences were not significant.

### 2.2. Analysis of Serological and Fecal Parameters across Follow-Up Time

All the serological and fecal parameters were analyzed prior to the third (T3) and the fifth (T5) infusions and compared to the baseline (T0, Table 2). While the levels of vitamin D and 90K were unchanged across time, CRP and FCP decreased significantly after each infusion, and the iron levels significantly increased. The same trend was observed after the data were stratified for disease (Appendix A).

### 2.3. Effect of Circulating Infliximab across Follow-Up Time

Prior to the third (T3) and the fifth (T5) IFX infusions, a subgroup of patients showed the presence of infliximab trough levels (ITLs), measured by ELISA assays (Table 3). While in almost all the T3 samples (43 of 48) there were detectable ITLs, they were progressively lost up to the fifth infusion, when ITLs were detected only in 19 of the T5 samples. This loss of detectable ITLs is independent of the type of disease, being evident in both the CD and UC patients. We then assessed the serum levels of 90K, CRP, and FCP in a group of patients with detectable ITLs, measured at the fifth infusion (“Yes” group), compared to a group with no ITLs in the serum (“No” group).

We detected lower 90K levels in the samples with circulating ITLs (Figure 1A); the CRP level was significantly lower in the presence of ITLs in the T5 samples (Figure 1B); instead, the FCP level was significantly lower from the first to the fifth infusion, with and without ITLs (Figure 1C). These differences were observed also after the samples were stratified for disease (Appendix A).

### 2.4. Circulating Antibodies to Infliximab Affect Serum Levels of Inflammatory Markers 

A sub-group of patients undergoing IFX infusion developed antibodies to infliximab (ATIs) during progressive infusions, thus becoming non-responders; this condition often leads to treatment failure and the physician’s decision to switch drug.

Table 4 and Figure 2 show the inflammatory markers measured at T3 and T5, in relation to the development of ATIs.

As expected, the number of patients who developed ATIs was higher at T5 (18 of 47) than T3 (only 6), in particular in CD patients prior to the fifth IFX infusion. As for ITLs, we stratified our cohort into two groups (“Yes” and “No”), based on the presence of detectable ATIs measured in the serum at the fifth infusion.

90K and CRP serum levels were higher in patients with ATIs (Figure 2A,B), regardless of the number of infusions. Interestingly, the baseline level of 90K was significantly higher in patients who developed ATIs, and this difference was retained across time (Table 4 and Figure 3A). On the contrary, the FCP level was progressively lower, independently of the presence of ATIs, and this difference was significant (Figure 2C).

The same differences were also retained when the samples were stratified for disease, especially for the CD samples (Appendix A and Figure 3B).

At baseline, the levels of 90K and CRP were significantly higher in CD patients who developed ATI at the fifth infusion, while the differences were non-significant for UC (Appendix A and Figure 3C). We can conclude that in the CD patients, 90K acts as a predictor of the secondary loss of response. 

### 2.5. Correlation and Association between 90K and Classical Inflammatory Markers

We then performed a Spearman correlation test between the concentrations of circulating 90K and CRP, and circulating 90K and FCP, measured across follow-up time, in the total cohort.

Figure 4 shows that there is a significant positive correlation between 90K and CRP for T0 samples, but this correlation is lost over time.

No correlation was observed, however, between serum levels of 90K and FCP at each time point, as shown in Figure 5.

Finally, Figure 6 shows the association between the 90K concentration measured at baseline and the development of ATIs, at the fifth infusion. In the univariate model, and in the model adjusted for gender, the association was significant (OR = 1.24, 1.04 to 1.47 95% C.I., and OR = 1.24, 1.04 to 1.48 95% C.I., respectively), further confirming the correlation between these two biomarkers.

## 3. Discussions

In this study, we evaluated serum 90K as a new circulating marker in IBD patients undergoing IFX therapy. 90K was compared to other routinely used biomarkers: CRP, Iron, Ferritin, 25OH-Vitamin D, and FCP. In addition, we related the circulating levels of IFX (ITLs) and antibodies to infliximab (ATIs), to other biomarkers. 

There was a high level of heterogeneity in terms of age and gender in our sample cohort. The heterogeneity in our cohort is also reflected in the serological data measured at baseline.

In our cohort, we found higher 90K serum levels at baseline in subjects who developed ATIs during treatment, this finding being significant for the CD patients but not significant for the UC patients. Recently, it has been reported that CD patients, particularly those in the active phase, had significantly higher levels of 90K than control subjects and UC patients [45]. The high 90K levels in the CD patients, rather than the UC patients, suggests that 90K might be involved in the clinical manifestation of CD, and may therefore be a specific biomarker to monitor CD disease activity [45].

Among other inflammatory markers, FCP has proven to be the best biomarker for monitoring mucosal response to IFX therapy [53,54,55]. In our study we found that, although patients had high levels of ATIs, FCP decreased over the course of infusions. This probably occurs because we cannot distinguish whether the measured ATIs are neutralizing antibodies or not. Additionally, we limited our analysis to the fifth infusion, so we cannot predict whether FCP levels would have increased in subsequent infusions, concurrently with antibodies’ development.

CRP levels were significantly lower in serum samples with detectable IFX levels, particularly in patients with CD. Patients who developed ATIs were considered non-responders, and showed high serum levels of CRP and 90K. Several studies in recent years have shown a correlation between high serum CRP levels and a loss of response to anti-TNF treatment [20,32,33,34,35]. Interestingly, we found that non-responder patients showed higher 90K serum levels at baseline than responders. Moreover, this difference was maintained across treatment. We found a positive correlation between CRP and 90K at baseline. Although an association between these biomarkers was previously demonstrated in Behçet’s disease [56], our finding is in contrast to the study by Cibor et al. [45], probably because patients included in that study underwent different treatment approaches, and only a small number of patients underwent biological therapy.

Our study presents some limitations, in particular the single center cohort and the small sample size. In the future, it will be of interest to increase the patient cohort sizes, in order to address the question whether the non-significant results are a reflection of the small size of the cohort or are closely related to biological differences between UC and CD. Furthermore, we limited our analysis to serum samples only, as our study aimed to search for non-invasive biomarkers; 90K epithelial expression data should add significance to our preliminary results.

Nonetheless, this is the first study to relate the 90K serum level with the monitoring of IFX therapy, by concomitantly determining ITLs and ATIs. In fact, although a preliminary finding, in our IBD cohort the relationship between 90K baseline levels and ATIs was observed to be predictive of the loss of response to IFX therapy.

## 4. Materials and Methods

### 4.1. Ethical Considerations

The study protocols on human subjects were conducted in accordance with the principles of the Declaration of Helsinki (1964) and approved by the local Ethics Committee, “Comitato Etico I.R.C.C.S.-Istituto Tumori “Giovanni Paolo II”-Bari” (protocol code Number 38 on 18 May 2017).

### 4.2. Study Population

In total, 48 IBD patients were enrolled at the IBD Unit of the National Institute of Gastroenterology “Saverio De Bellis”, IRCCS (Castellana Grotte, BA, Italy) and underwent IFX infusions, from February 2017 to December 2018. Among these, 30 (62.5%) were CD patients and 18 (37.5%) were UC patients (Appendix A).

Written informed consent was obtained at recruitment from all subjects, following the principles of the Declaration of Helsinki (1964). There was no other pharmacological intervention in the study, since it was observational. 

The anti-TNF (Remicade^®^, Janssen, Beerse, Belgium or the biosimilar Remsima^®^, Celltrion, Incheon, Republic of Korea) drug was administered intravenously at a dosage of 5 mg/kg body weight; the interval between infusions varied according to clinical decisions although, in any case, temporal variations between one infusion and the next were minimal. No distinction was made between the original drug and the biosimilar, as previous studies have demonstrated no difference in terms of efficacy, safety, and immunogenicity between the original and the biosimilar [44].

The inclusion criteria were: patients scheduled for treatment with anti-TNF therapy, to be performed with either Remicade^®^ or the biosimilar Remsima^®^, and no previous treatment with anti-TNF therapy. Exclusion criteria were: comorbidities (assessed with the Charlson Comorbidity Index), ongoing immunosuppressive or immunomodulatory therapy, a malignant neoplasm in the last 10 years, pregnancy or breastfeeding, and the need for artificial nutrition.

Patients underwent fasting venous sampling prior to the first (T0), third (T3) and fifth (T5) infusions of IFX biological therapy. The samples were collected in 5 ml Vacutainers Gel Tubes (Becton Dickinson, Franklin Lakes, NJ, USA).

### 4.3. Serological Tests

After blood clot formation and centrifugation, each serum sample was tested for the determination of C-reactive protein (CRP), iron, and ferritin through an automated analytical integrated system (Beckman Coulter DXi and DXc 600, Brea, CA, USA); and 25OH-vitamin D through an automated chemiluminescent platform (DiaSorin Liaison XL, Saluggia, VC, Italy).

### 4.4. Quantification of Fecal Calprotectin (FCP)

Fecal samples were collected at home or in hospital. When collected at home, patients were instructed to refrigerate their fecal samples (2–8 °C). The fecal samples were stored in aliquots at −20 °C or below until analysis using an automated chemiluminescent platform (DiaSorin Liaison XL, Saluggia, VC, Italy). 

### 4.5. ELISA Assays

Infliximab trough levels (ITLs), and antibodies to infliximab (ATIs), were assessed simultaneously by ELISA assay (LISA Tracker Duo IFX, Theradiag, Marne-la-Vallée, France) using a serum aliquot stored at −80 °C until assay. The cut-offs for therapeutic ITLs and detectable ATIs were established at 3 μg/mL and 10 ng/mL, respectively, according to the manufacturer’s instructions. The ITLs and ATIs levels were determined at the third and fifth infusions. The concentration of serum 90K glycoprotein was determined by ELISA assay, according to the manufacturer’s instructions (Human s90K/Mac-2 BP, Invitrogen, Waltham, MA, USA).

### 4.6. Statistical Analysis

Data are reported as Mean ± Standard Deviations (M ± SD) for continuous measures, and frequency and percentages (%) for all categorical variables.

To test the associations at each time point between infusion (ITLs) and antibodies (ATIs), a Chi-square test or Fisher’s exact test was used, when necessary, for categorical variables. For non-normally distributed variables, the Wilcoxon rank-sum (Mann–Whitney) test was used.

To evaluate the trend of blood parameters over time, the test for trend was used. 

The Spearman rank correlation coefficient was used to test the strength and direction of association between two variables examined (i.e., between 90K and CRP, and 90K and FCP). Analysis of Variance (ANOVA) for repeated measures, in continuous variables, was used to evaluate the ATIs development (Yes/No) and the time effect referred to infusions. The test for trend across infusions was used to examine variations over time for the CRP and 90K variables.

To evaluate the association between ATIs at the fifth infusion (Yes/No) and 90K at baseline, logistic regression was used and the odds ratio (OR) reported as estimation.

When testing the hypothesis of a significant association, the *p*-value was < 0.05, two-tailed for all analyses.

All statistical computations were made using the STATA, Statistical Software, StataCorp 2019, Release 16 (StataCorp LLC, College Station, TX, USA), and the RStudio software (“Prairie Trillium” Release).

## 5. Conclusions

The significant positive correlation found between 90K and CRP measured at baseline indicates that 90K could be considered as a new predictive biomarker of response to IFX therapy. In short, a patient with high 90K serum levels would not be a good candidate for IFX therapy. 

Further investigations are needed to evaluate the 90K trend in subsequent infliximab infusions in IBD patients, since it is known that non-responder patients will increase in parallel with the number of infusions.

The results of our preliminary study pave the way to a new, larger multicenter study, performing prospective monitoring of 90K serum levels across all anti-TNF therapy. A larger sample size will increase the statistical power and further emphasize the role of the 90K serum level as a biomarker in UC and CD. In addition, quantification of 90K protein expression in cellular subcompartments on immunohistochemical colon sections, along with gene expression analysis, will add more information about the role of 90K in IBD-associated inflammation, especially in relation to the development of ATIs.

Future works will aim to clarify the role of 90K in IBD, and the mechanism that contributes to the development of antibodies to anti-TNF therapy. 

## Figures and Tables

**Figure 1 ijms-24-03955-f001:**
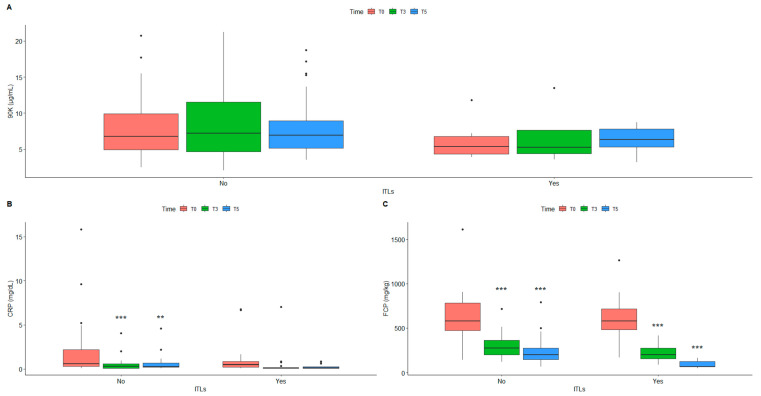
Comparison between serum level of 90K (**A**), CRP (**B**) and FCP (**C**) in patients with ITLs (No/Yes) measured at baseline (T0), before the 3rd (T3) and 5th infusion (T5). ** *p* ≤ 0.01; *** *p* ≤ 0.001 (No vs. Yes for each timing). Number of patients: total cohort = 47; No = 28; Yes = 19. CRP: C-reactive protein; FCP: fecal calprotectin; ITLs: infliximab trough levels.

**Figure 2 ijms-24-03955-f002:**
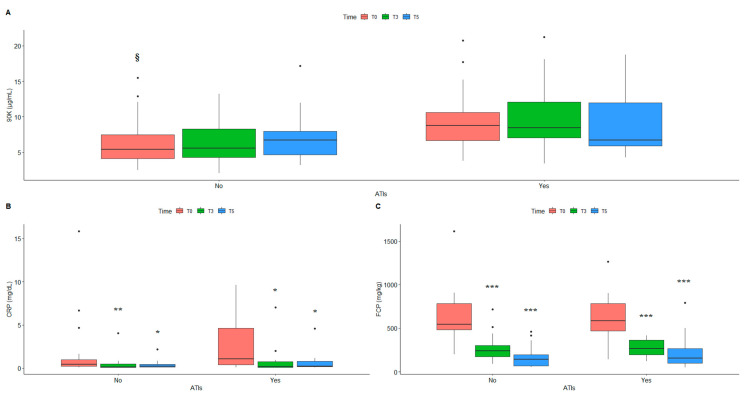
Comparison between serum levels of 90K (**A**), CRP (**B**) and FCP (**C**) in patients with ATIs (No/Yes) measured at baseline (T0), before the 3rd (T3) and 5th infusion (T5). * *p* ≤ 0.5; ** *p* ≤ 0.01; *** *p* ≤ 0.001 (No vs. Yes for each timing); § *p* ≤ 0.5 (T0 No vs. T0 Yes). Number of patients: total cohort = 47; No = 29; Yes = 18. CRP: C-reactive protein; FCP: fecal calprotectin; ATIs: antibodies to infliximab.

**Figure 3 ijms-24-03955-f003:**
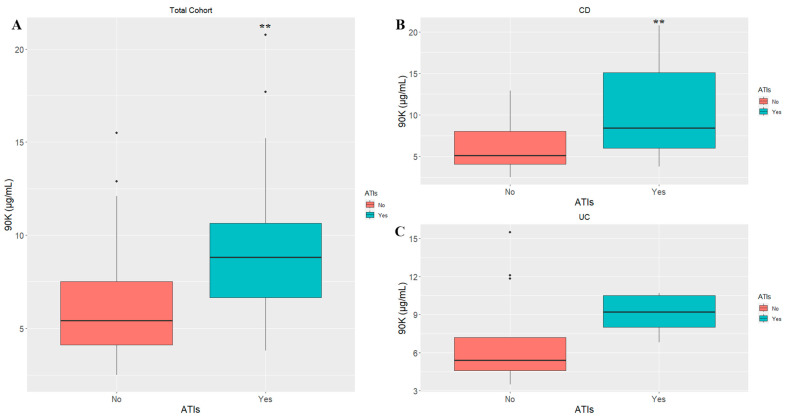
Comparison between serum levels of 90K measured at baseline, in IBD patients who developed ATIs (No/Yes). (**A**): Total Cohort (N = 48); (**B**): CD patients (N = 30); (**C**): UC patients (N = 18). ** *p* ≤ 0.01 (No vs. Yes). ATIs: antibodies to infliximab; CD: Crohn’s disease; UC: ulcerative colitis.

**Figure 4 ijms-24-03955-f004:**
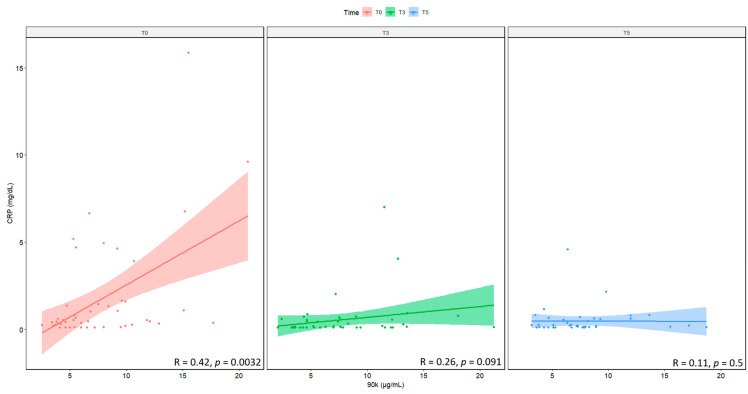
Correlation between serum levels of 90K and CRP, measured at baseline (T0), prior to the 3rd (T3) and the 5th (T5) infusion. N = 48. CRP: C-reactive protein.

**Figure 5 ijms-24-03955-f005:**
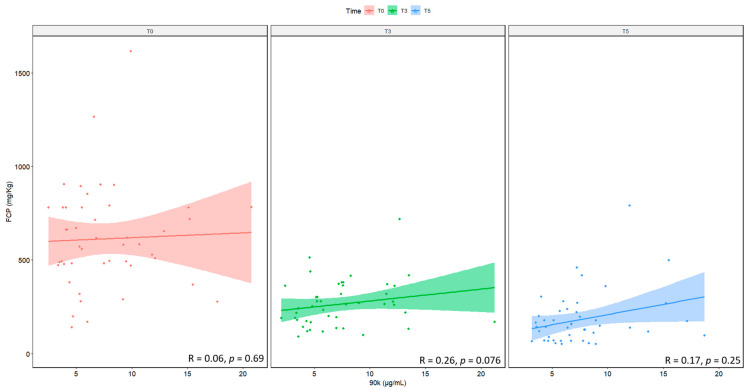
Correlation between serum levels of 90K and FCP, measured at baseline (T0), prior to the 3rd (T3) and the 5th (T5) infusion. N = 48. FCP: fecal calprotectin.

**Figure 6 ijms-24-03955-f006:**
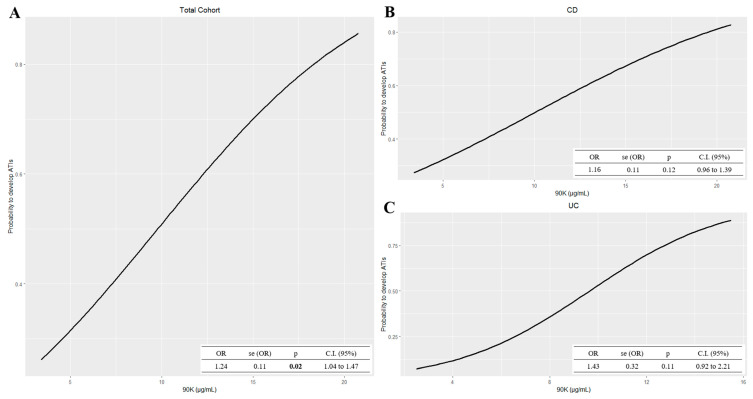
Plots of univariate logistic regression models of 90K at baseline on ATIs development, in (**A**) the total cohort (N = 48), (**B**) CD (N = 30), and (**C**) UC patients (N = 18). Black continuous lines show logistic model of the probability of developing ATIs (y-axis) in relation to increased 90K levels (x-axis). ATIs: antibodies to infliximab; CD: Crohn’s disease; UC: ulcerative colitis.

**Table 1 ijms-24-03955-t001:** Clinicopathologic features of enrolled patients (N = 48).

Parameters †	Total Cohort Mean ± SD (N = 48)	CD Mean ± SD (N = 30)	UC Mean ± SD (N = 18)
Gender N (%)			
Male	30 (62.50)	15 (50.0)	15 (83.33)
Female	18 (37.50)	15 (50.0)	3 (16.67)
Age (years)	45.87 ± 21.74	41.44 ± 17.09	53.78 ± 27.14
Smokers N (%)	5 (16.13)	5 (23.81)	0 (0.00)
Mayo Full score (UC) (%)			
1-Mild disease		--	0.00
2-Moderate disease		--	17.65
3-Severe disease		--	82.35
Location of UC disease patients—Montreal Classification (%)			
E1-Ulcerative (proctitis)		--	0.00
E2-Left sided UC (distal UC)		--	26.32
E3-Extensive UC (pancolitis)		--	73.68
Disease Behavior (B)—Montreal Classification (%)			
B1-Inflammatory (non stricturing-non penetrating)		81.48	--
B2-Stricturing		14.81	--
B3-Penetrating		3.70	--
Perianal CD		0.00	--
Location of CD disease patients (L)—Montreal Classification (%)			
L1-Ileal	--	33.33	--
L2-Colic	--	11.11	--
L3-Ileocolic	--	55.56	--
L4-Upper GI	--	0.00	--
HBI (Harvey–Bradshaw Index)		15.52 ± 4.40	--
Serological and fecal data at baseline			
25OH-Vitamin D (ng/mL)	25.03 ± 11.12	24.08 ± 9.72	26.70 ± 13.39
90K (µg/mL)	7.71 ± 4.11	7.78 ± 4.56	7.59 ± 3.36
CRP (mg/dL)	1.72 ± 2.99	1.55 ± 2.31	2.01 ± 3.93
Iron (µg%)	57.03 ± 55.86	62.82 ± 66.67	46.82 ± 27.17
Ferritin (ng/mL)	62.12 ± 57.95	64.95 ± 61.96	57.13 ± 51.53
FCP (mg/kg)	613.67 ± 272.75	622.34 ± 303.66	598.88 ± 217.95

† As Mean and Standard Deviation (M ± SD) for continuous variables, and percentage (%) for categorical. CD: Crohn’s disease; UC: Ulcerative colitis; GI: gastrointestinal; HBI: Harvey–Bradshaw index; CRP: C-reactive protein; FCP: fecal calprotectin.

**Table 2 ijms-24-03955-t002:** Comparison between infusion cycles and baseline, and trend for the total cohort (N = 47). Statistically significant differences (*p*-value ≤ 0.05) are indicated in bold.

Parameters †		Infusion Cycles ‡				
	T0 (a)	T3 (b)	*p* § (b vs. a)	T5 (c)	*p* § (c vs. a)	*p* for Trend ¶
25OH-vitamin D (ng/mL)	25.03 ± 11.12	25.36 ± 12.08	0.99	24.32 ± 9.60	0.62	0.79
90K (µg/mL)	7.71 ± 4.11	7.81 ± 4.13	0.11	7.61 ± 3.66	0.55	0.90
CRP (mg/dL)	1.72 ± 2.99	0.56 ± 1.18	**0.001**	0.48 ± 0.78	**<0.0001**	**0.04**
Iron (µg%)	57.03 ± 55.86	73.57 ± 62.81	**0.001**	80.92 ± 49.32	**0.01**	**0.05**
Ferritin (ng/mL)	62.12 ± 57.95	45.77 ± 52.93	**0.01**	74.17 ± 118.95	0.87	0.55
FCP(mg/kg)	613.67 ± 272.75	265.50 ± 122.82	**<0.0001**	181.11 ± 140.95	**<0.0001**	**<0.0001**

† As Mean and Standard Deviation (M ± SD). ‡ Baseline (T0), prior to the 3rd (T3) and 5th (T5) infusion. § Wilcoxon signed-rank test; ¶ Test for trend. CRP: C-reactive protein; FCP: fecal calprotectin.

**Table 3 ijms-24-03955-t003:** Analysis of variance for repeated measures, at Baseline, 3rd and 5th infusion, for single measured parameters in patients with ITLs at T5 (No/Yes; Number of patients: total cohort = 47; No = 28; Yes = 19). Statistically significant differences (*p*-value ≤ 0.05) are indicated in bold.

Parameters †		Infusion Cycles ‡			Effects *p* §		Contrasts *p* ¶ vs. (a)	
	ITLs	T0 (a)	T3 (b)	T5 (c)	ITLs	Time	(b)	(c)
90K (µg/mL)	--	7.71 ± 4.11	7.81 ± 4.13	7.61 ± 3.66	0.45	0.98	--	--
	No	8.14 ± 4.06	8.11 ± 4.04	7.86 ± 3.62	--	--	0.96	0.93
	Yes	7.22 ± 4.30	7.37 ± 4.46	7.25 ± 3.78	--	--	0.82	0.96
	*p*-value ¥	0.45						
CRP (mg/dL)	--	1.72 ± 2.99	0.56 ± 1.18	0.48 ± 0.78	0.34	**0.001**	--	--
	No	2.16 ± 3.51	0.55 ± 0.84	0.65 ± 0.99	--	--	**<0.001**	**0.003**
	Yes	1.15 ± 2.00	0.57 ± 1.57	0.27 ± 0.26	--	--	0.09	0.98
	*p*-value ¥	0.09						
FCP (mg/kg)	--	613.67 ± 272.75	265.50 ± 122.82	181.11 ± 140.95	**0.02**	**<0.0001**	--	--
	No	628.92 ± 292.87	293.50 ± 135.19	246.23 ± 156.43	--	--	**<0.001**	**<0.001**
	Yes	589.79 ± 256.38	221.10 ± 91.50	94.89 ± 39.20	--	--	**<0.001**	**<0.001**
	*p*-value ¥	0.45						

† As Mean and Standard Deviation (M ± SD). ‡ Baseline (T0), prior to the 3rd (T3) and 5th (T5) infusion. ¥ ITLs at 5th (T5) infusion; § ANOVA for repeated measures; ¶ Contrasts of marginal linear predictions. ITLs: infliximab trough levels; CRP: C-reactive protein; FCP: fecal calprotectin.

**Table 4 ijms-24-03955-t004:** Analysis of variance for repeated measures, at Baseline, 3rd and 5th infusion, for single measured parameters in the presence of ATIs at T5 (No/Yes) in IBD patients (Number of patients: total cohort = 47; No = 29; Yes = 18). Statistically significant differences (*p*-value ≤ 0.05) are indicated in bold.

Parameters †		Infusion Cycles ‡			Effects *p* ^§^		Contrasts *p* ¶ vs. (a)	
	ATIs	T0 (a)	T3 (b)	T5 (c)	ATIs	Time	(b)	(c)
90K (µg/mL)	--	7.71 ± 4.11	7.81 ± 4.13	7.61 ± 3.66	**0.007**	0.91	--	--
	Yes	9.76 ± 4.65	9.80 ± 4.80	8.88 ± 4.45	--	--	0.96	0.37
	No	6.53 ± 3.29	6.58 ± 3.26	6.86 ± 2.93	--	--	0.92	0.53
	*p*-value ¥	**0.005**						
CRP (mg/dL)	--	1.72 ± 2.99	0.56 ± 1.18	0.48 ± 0.78	0.21	**0.0005**	--	--
	Yes	2.33 ± 2.81	0.80 ± 1.67	0.74 ± 1.20	--	--	**0.007**	**0.02**
	No	1.40 ± 3.13	0.41 ± 0.76	0.36 ± 0.43	--	--	**0.02**	**0.02**
	*p*-value ¥	0.12						
FCP (mg/kg)	--	613.67 ± 272.75	265.50 ± 122.82	181.11 ± 140.95	0.75	**<0.0001**	--	--
	Yes	600.65 ± 285.97	267.35 ± 96.72	213.70 ± 184.64	--	--	**<0.001**	**<0.001**
	No	619.53 ± 274.34	260.25 ± 138.10	163.28 ± 108.61	--	--	**<0.001**	**<0.001**
	*p*-value ¥	0.75						

† As Mean and Standard Deviation (M ± SD). ‡ Baseline (T0), prior to the 3rd (T3) and 5th (T5) infusion. ¥ ITLs at 5th (T5) infusion; § ANOVA for repeated measures; ¶ Contrasts of marginal linear predictions. ATIs: antibodies to infliximab; CRP: C-reactive protein; FCP: fecal calprotectin.

## Data Availability

The raw data supporting the conclusions of this article will be made available by the authors, without undue reservation.

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
