# Peer review of "90K/Mac-2 BP Is a New Predictive Biomarker of Response to Infliximab Therapy in IBD Patients"

_ijms, 2023, doi:10.3390/ijms24043955_

Round 1

Reviewer 1 Report

In this manuscript, Pasqua Letizia Pesole et al. reported the potential of 90K/Mac-2 BP for the predictive biomarker of response to infliximab therapy. Although this study's results help physicians choose biologics for patients with IBD, more information should be described for the conclusion.

Specific recommendations for revision-a) major

Although the authors divided the patients into two groups, "Yes" or "No," for analysis with ATI and ITL, the definitions of these groups should be described in the manuscript, such as the cut-off values of concentrations or the timing of measurements. The numbers for these groups should be added to each Table.

The information on concomitant treatments of the patients is critical for this study, particularly azathioprine which could prevent loss of response to IFX.

Specific recommendations for revision-b) minor

The first "I" of Infliximab is lowercase letter.

The abbreviations for the terms in Tables are required.

Anti-IFX in Table S6 and S7 could be ATI.

Table 1 and Table S1 could be combined, and the format of Table S1 is better for the readers.

In figure 1, the text should be larger, and lines should be thicker. The bars are required to indicate the significant differences.

In Table 2, S2, and S3, "b vs a" under T3 and "c vs a" under T4 may be b and c, respectively, and they should be listed under each p-value.

#22 is not cited in the references.

Reviewer 2 Report

Review for the manuscript 90K/Mac-2 BP is a new predictive biomarker of response to the 2 Infliximab therapy in IBD patients submitted to IJMS.

Dear Editor, thank you for the opportunity to review this manuscript. After careful evaluation, I suggest some modifications before it can be published.

Overall comments:

            This is a very interesting manuscript.

ABSTRACT

            In this section, we may find “Abstract: Inflammatory bowel diseases, comprising Crohn’s disease and Ulcerative Colitis, are multifactorial disorders characterized by a chronic inflammatory status with secretion of cytokines and immune mediators. In the last decades, biologics targeting pro-inflammatory cytokines, such as Infliximab, have been used in the treatment of IBD patients,…”. I suggest changing to “Abstract: Inflammatory Bowel Diseases (IBD), comprising Crohn’s disease and Ulcerative Colitis, are multifactorial disorders characterized by a chronic inflammatory status with the secretion of cytokines and immune mediators. In the last decades, biologics targeting pro-inflammatory cytokines, such as Infliximab, have been used in the treatment of IBD patients.”
Please, do not use an abbreviation without a definition.

            In this section, I also suggest reducing the Introduction and improving the description of the Results.

KEYWORDS

            Keywords are: IBD; 90K/Mac-2 BP; Galectin-3 binding protein; Infliximab. I suggest “Keywords: Inflammatory bowel diseases; 90K/Mac-2 BP; Galectin-3 binding protein; Infliximab”

INTRODUCTION

            This section is adequately described. However, there are a plethora of new articles on IBD pathophysiology, epidemiology, and general aspects. Please, include more references published in 2022 in this section. Please consult PUBMED.com.

METHODS

            This section was adequately described. However, we can find in lines 236-238: “The study protocols on human subjects were conducted in accordance with the principles of the Declaration of Helsinki and approved by the local Ethics Committee (Number 38 National Institute of Gastroenterology “S. De Bellis”)”. Please, include the year for the Declaration of Helsinki.

In line 256: exclusion criteria were “comorbidities with 256 other diseases (assessed with the Charlson Comorbidity Index), ongoing immunosuppressive or immunomodulatory therapy, a malignant neoplasm in the last 10 years, pregnancy or breastfeeding and the need for artificial nutrition”. Were patients with recent surgical procedures part of these exclusion criteria? And the age of the participants?

            In line 239, Study population: was there a sample calculation for the selection of the patients? Please, clarify the reason for the inclusion of 48 participants.

                        I suggest the inclusion of a flowchart showing the selection of the patients, the number of CD and UC patients, and the dropouts.

RESULTS

            Please, define the abbreviations in the legend of all Figures.

            In the legend of table 1 we can see “† As Mean and Standard Deviation (M±SD) for continuous variables, and percentage (%) for categorical. Abbreviations: CD, Crohn's disease; UC, Ulcerative colitis; HBI, Harvey-Bradshaw index”. Please, change for “† As Mean and Standard Deviation (M±SD) for continuous variables, and percentage (%) for categorical. CD, Crohn's disease; UC, Ulcerative colitis; HBI, Harvey-Bradshaw index”.

            Still in table 1: I wonder why alcohol use was not considered.

            Please, see that in the title of table 2 there is 2 periods (Table 2. Comparison between infusion cycles and baseline, and trend for total cohort (N=48). Statistically significant differences (p-value ≤ 0.05) are underlined in bold. .)

            In the legend of table 2: please include the definitions for the Abbreviations.

            In lines 101-103, we can read “Table S1 shows the baseline characteristic of the enrolled patients, stratified by disease. In UC patients we detected a higher CRP level and lower circulating iron, ferritin and FCP…” Please, note that in this place it was the first time that “FCP” was used without the definition. The definition is only first mentioned in the line 269. Please, correct this.

            The first parameter of Table 2 is “25OH-Vitamin D (ng/ml)”. Please, correct ml to mL (as the authors used for the other parameters).

            Please include abbreviations in the legend of table 3. The same is true for table 4.

            The words and numbers in Figure 2 and Figure 3 are too small. I suggest increasing the size and using a darker color.

DISCUSSION

            This section is adequate. I have only one question:

In lines 213-214 we find “Treatment with 90K of PBMCs induces the secretion of various cytokines such as IL 6, GM-CSF and TNF”. This is for all the entire family of TNF? Or the authors refer to TNF-alpha?

Please, include the limitations of this study.

CONCLUSION

            This section is adequate.

REFERENCES

            As pointed out above, please include more references published in 2021 and 2022.

Reviewer 3 Report

The authors performed a lot of analyses in this study. However, the manuscript lacks consistency, accuracy and it appears confusing. Major revision of the English language is required. Manuscript appears difficult to follow. Please see below my comments/questions:

1. Abstract.

a. Please add: period of the study, number of patients, how many with UC/CD, design- retrospective, prospective; multicenter/single center; 

b. Aim of the study: please replace “correlate” with “analyze the relationship” (or whatever you think is better); “correlate” is not appropriate in this instance, as a priori, we do not know whether a correlation does exist or not. Also, it should be mentioned "90K serum level at baseline". Please insert or clarify.

c. Please mention the duration of follow-up. When did you measure anti-Infliximab antibodies? it is needed in the Abstract.

d. The authors wrote: "Furthermore, 90K serum level determination before the first infliximab infusion, would help the  physicians in the choice of the biologics for treatment of IBD patients". Please specify what happens then. Would you not give IFX, just because of this marker? This is crucial for physicians. From the main text, I am not convinced at all.

e. Generally, the Abstract is very flimsy, whilst it should be representative of the whole paper. Please include all required data and make it complete.

2. Keywords: It would be advisable to use other Keywords, not those belonging to the title. This would increase the likelihood of the paper being found by readers. The importance of Keywords is to improve indexing. Only “Galectin-3 binding protein” is not part of the title.

3. Introduction.

This paragraph should be based on more recent references. They are very old, while a plethora of very recent excellent manuscripts exist. This is why, here and there, the info is not complete. Also, please expand more on therapeutic drug monitoring (proactive… reactive…), based on recent data. And about CRP – please new data/references. Please refine your aim (see above) and also remove “confirm” – when you begin a study, you do not know whether you can confirm it or not. Also, aim should match the one in the Abstract.

4. Results

a. Lines 88-89: You mentioned that “48 IBD patients (30 CD and 18 UC) have been enrolled on the day of the first IFX infusion; 47 patients completed the study and one dropout was registered.” Therefore, since 47 patients completed the study, why at T3 and T5, there are still 48? Very confusing.

b. Please use the following sentences in Discussion: “There was a high grade of heterogeneity in terms of age and gender in our sample cohort. The heterogeneity in our cohort is also reflected in the serological data measured at baseline.

c. Please mention what score was used to determine activity: “All the UC patients had a moderate to severe disease”. It is mentioned in Table 1, but for CD, you mentioned in the whole text the Harvey-Bradshaw Index.

d. When mentioning behavior, please indicate also “p” (perianal). It is not mentioned in Table 1, either.

e. In the sentence – “According to the Montreal Classification, 33.33% of CD patients had an ileal disease, 11.11 % a colic phenotype and 55.56% an ileocolic disease.”, please use then the respective L (location). What about L4? I see later that Table 1 also does not mention L4. Please add.

f. Same for behavior (lines 95-97). If you do not want to use “B”, then correct “structuring”, as it is not correct.

g. Please define abbreviation “FCP” before using it. Fecal calprotectin was not defined before.

h. Table 1 – is different from the main text. Line 94: “All the UC patients had a moderate to severe disease”. Please replace “to” with “or”, according to Table 1.

i. Table 1 – E1 means proctitis. It is not mentioned. please add.

k. Line 119 – please give details about IFX trough levels.

l. Please explain how come: “the FCP level was progressively lower, independently from the presence of ATIs, and this difference was significant”. (comments in Discussion).

m. In Abstract, the authors wrote “In our IBD cohort, significantly higher 90K serum levels have been found at baseline, in patients who developed anti-Infliximab antibodies, thus becoming non responder, compared to responder patients”. However, in Results, this is valid only for the CD patients: “At baseline, the level of 90K and CRP was significantly higher in CD patients who developed ATI at 5th infusion, while the differences were non significant for UC”. Please clarify.

5. Discussion: This paragraph contains a lot of info that is just theory, some of the data being redundant with Introduction. Also, there is a lot of data not related to this manuscript and therefore not needed in this paragraph. The authors should start with their results, interpretation, comments, hypotheses raised and so on. In total, this paragraph should be re-written. Limitations lack completely. Very small number of patients should be included and many more. Strength? Also, what are the practical implications of your study? What proper directions for further research?

6. Materials and Methods: I understand this study was finished in 2018. But, this does not justify the ancient references.

Statistical analysis: Multivariate logistic regression analysis should be performed in order to adjust for potential confounders. The results cannot be trusted and I would not just avoid IFX in a patient with high 90K serum level at baseline.

7. Conclusion has no meaning. Also, future directions were inserted here, but just vaguely. How do the authors now that “Future works will clarify the role of 90K in IBD, and the molecular mechanism that contribute to the developments of antibodies to anti-TNF therapy”?

8. References: 22 is absent.

At least 9 out of 40 References include some of the authors of the present study (at least 22.5%) and some of them are not needed.

Generally, references are very old. Please update.

Round 2

Reviewer 1 Report

Thank you for your revision.

Reviewer 3 Report

I am very glad that the Authors considered the reviewers’ suggestions/comments and they worked really hard to greatly improve their manuscript. They realized what was inconsistent or wrong and they corrected existent info/added missing data. Now, the manuscript appears logical, scientifically correct, reliable and everything makes sense. This new version of the paper will be useful for our community of GIs. I strongly support its publication in the present form.